# Cage Effect under Photolysis in Polymer Matrices

**Igor V. Khudyakov** [1,*] **, Peter P. Levin** [2,3] **and Aleksei F. Efremkin** [3,†]

1   Department of Chemistry, Columbia University, New York, NY 10027, USA
2   Emanuel Institute of Biochemical Physics, Russian Academy of Sciences, 119334 Moscow, Russia; levinp@mail.ru
3   Semenov Institute of Chemical Physics, Russian Academy of Sciences, 119991 Moscow, Russia; aefremkin@yandex.ru
*   Correspondence: startatj@gmail.com
†   Dr. A.F. Efremkin passed away on 12 January 2019.

**Abstract:** Photoinduced elementary reactions of low-MW compounds in polymers is an area of active research. Cured organic polymer coatings often undergo photodegradation by free-radical paths. Besides practical importance, such studies teach how the polymer environment controls elementary free-radical reactions. Presented here is a review of recent literature which reports such studies by product analysis and by a time-resolve technique of photochemical reaction inside the cage of a polymer and in the bulk of a polymer. It was established that application of moderate external magnetic field allows the control of the kinetics of free radicals in elastomers. Preheating and stretching of elastomers affect reactivity of photoproduced radicals.

**Keywords:** cage effect; geminate recombination; F- and G-radical pairs; polymers; photodissociation; photoreduction; magnetic field effect

## 1. Introduction

A description of cage effect usually starts with the first publication on this subject. The term cage effect was suggested in 1934 by Frank and Rabinowitsch [1]. Originally, it meant an effect of the liquid (condensed) phase on quantum yield of photolysis of molecules studied initially in the gas phase. That quantum yield significantly decreases when running the reaction in the liquid phase compared to the gas phase. The essence of cage effect is that in liquid, contrary to gas phase, molecules undergo not one but a series of subsequent collisions. It is recommended to use the term contact of molecules in the liquid phase. A series of contacts leads to an encounter of molecules in the liquid phase.

It is believed that photogenerated molecules are born in a certain cage of a solvent, which does not allow their immediate separation. Molecules undergo recombination into the starting or some other product in the cage. Concurrently photogenerated molecules separate from each other and react with each other or with other molecules like acceptors on the solvent bulk, as shown in Scheme 1.

In the most studied cases, the pair A . . . B and the pair $A_1$ . . . $B_1$ are free radicals. We will describe the case when A*, being in a triplet state, abstracts a hydrogen atom from a donor B (Scheme 1). Products of the cage recombination are not necessarily the same as the starting products.

Molecules/radicals A and B are generated at distance close to or somewhat larger than a sum of their van der Waals radii $\rho_A + \rho_B$. The latter sum is considered as a reaction radius $\rho_{12}$. In most cases, neutral molecules/radicals react in direct contact at a distance of $\rho_{12}$. It is not possible to determine the size of a cage and a time of cage existence (or the time of geminate recombination/dissociation). It is believed that a cage size ($r_c$, m) is of order of $\rho_{12}$, ($\rho_{12}$, m) and a cage process takes place during

$$\tau_c \text{ (s)} \approx \rho_{12}^2/D_{12}, \tag{1}$$

where $D_{12}$ (m$^2$/s) is a mutual diffusion coefficient, $D_{12} = D_A + D_B$ [2–4].

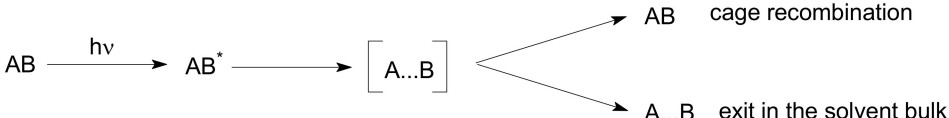

**Scheme 1.** Square brackets denote products of photoreactions are inside a solvent cage. Photoreactive molecules AB (top part) or A (bottom part) can be in a singlet or in a triplet state. The triplet state is formed after intersystem crossing in the excited singlet molecule.

A probability of reactions of molecules while they are in a cage with other molecules in solution is negligible, except with a huge (~1 M and higher) concentration of an added reagent-scavenger [5]. A second possible extreme case is a very high concentration of generated pairs. In such a case, the reagent may react not only with its counter reagent but with a reagent from a neighboring pair.

The notion of cage effect is widely used in the kinetics of elementary liquid phase and especially free-radical reactions. Cage effect value $\varphi$ is a fraction of photogenerated molecules (radicals) recombined within a cage to the total number of generated radicals. Cage escape value $e$ is a fraction of radicals escaped from the cage into the solvent bulk. Obviously:

$$\varphi + e = 1.0 \tag{2}$$

We will consider the formation of free radicals according to Scheme 1. Singlet pairs usually promptly react with each other, whereas triplet pairs react slower (see below). A caged pair of radicals is formed in the same spin state as its excited precursor, that is, in a singlet or in a triplet state. Interconversion between singlet and triplet cages is an additional factor influencing cage reactions:

$$^1[A\ldots B] \rightleftharpoons {}^3[A\ldots B] \tag{3}$$

On the terminology used in the literature: Caged radicals are often called radical pairs (RPs) of G-pairs. Radicals, which form a pair during a random encounter in the bulk of a solvent, are called free-radical pairs (F-pairs). Cage recombination is often called geminate recombination. For brevity, bimolecular reaction between free radicals is called recombination; it may be a disproportionation, in fact. Exit into the solvent bulk is often called cage escape.

Singlet (S)–triplet (T) interconversion of RPs (Equation (3)) during $\tau_c$ adds spin chemistry and magnetochemistry to the kinetics of elementary radical reactions in the condensed phase.

In this review article, we will consider reactions of neural free radicals in polymers in G- and F-pairs. Cage reactions are usually very fast due to the fact that the reagents are in proximity to each other. It is desirable to slow down cage reactions for direct observation of cage reaction kinetics and $\varphi$ in order not to interfere with reactions of photoexcited precursors of radicals.

(Viscous) polymers should be a convenient media for a study of the kinetics of such reactions.

Photodegradation of polymers often proceeds by the free-radical pathway, and primary chemical cage reactions essentially determine the sustainability of a polymer. Most of the organic coatings produced by photopolymerization have a residual low-MW photoinitiator (PI). That PI becomes one of the causes of photodegradation of the cured coatings outdoors. In particular, benzophenone (see below), is a common PI.

There are a number of review articles dedicated to the cage effect [2–8].

## 2. Cage Effect Values by Product Analysis

There is a multitude of individual commercial polymers, polymeric coatings with quite different properties. We will cite in this work several polymers which were used as a matrix for the study cage effect under photolysis of low-MW compounds dissolved or dispersed in these polymers. One approach is based on product analysis of photodissociation of organic compounds in polymers. It was successfully used by Weiss, Chesta, and their collaborators [8–13]. Norrish type I cleavage of 1-(4-alkylphenyl)-3-phenyl-2-propanone (ACOB) was used as a probe of polymer matrices:

The occurring reactions are presented in Scheme 2. ACOB under photoexcitation undergoes intersystem crossing with a formation of a triplet (T) state. The T-state dissociates with the formation of two radicals. The arylacetyl radical quickly dissociates with the creation of a triplet G-pair consisting of benzyl and substituted benzyl radicals. A fraction φ of all photogenerated radicals undergoes interconversion into singlet RP (Equation (3)) and recombines within a cage. In some special cases, T-pairs recombine in the T-state caused by an action of spin–orbit coupling (SOC) [3,7]. SOC, the interaction between spin and orbital angular momenta in the RP, can be relatively strong and allow spin S-T interconversion at the expense of a change in the orbital momentum [2]. Such cases can be met in RPs in special orbital configurations, in the long-lived pairs, pairs with a heavy atom [2,3,7].

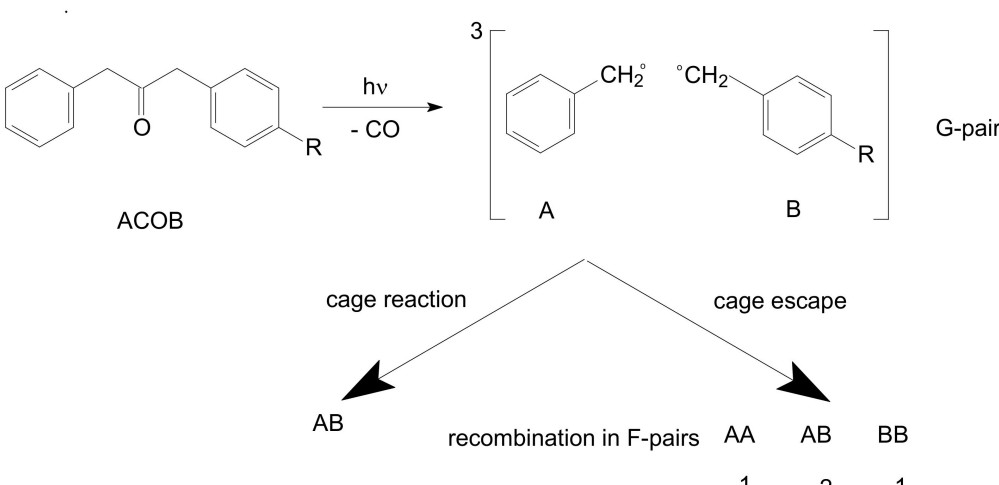

**Scheme 2.** Elementary chemical reactions during photolysis of substituted dibenzyl ketone.

An approach for investigation of elementary reactions presented in Scheme 2 was widely used for microheterogeneous media by Turro and collaborators [2].

The radicals not reacted in a cage exit into the polymer bulk and participate in random recombination in F-pairs. The ratio of concentrations of the products recombined in F-pairs is 1:2:1 (see Scheme 2). It is assumed that the reactivity of two radicals is equal, and distribution of products is determined only by probabilities of radicals meeting each other in the polymer bulk. The assumption of equal reactivity is probably correct in the case of 4-methylbenzyl radical (R = $CH_3$, Scheme 2) and benzyl radical. The assumption can be verified by product analysis. Concentrations of AA and BB should be equal. Under such assumptions for cage effect φ:

$$\varphi = \frac{[AB] - 2[AA]}{[AB] + 2[AA]} \tag{4}$$

Equation (4) is often presented in the literature in another form [8–11]:

$$\varphi = \frac{[AB] - [AA] - [BB]}{[AA] + [AB] + [BB]} \tag{5}$$

It is not surprising that Equation (5) gives distorted data when B has a structure essentially different from A. The radical 4-hexadecylbenzyl (B with R = $n$-$C_{16}H_{33}$, see Scheme 2) diffuses much slower than A, leading to violation of Equation (5). We believe that 4-hexadecylbenzyl radical has also lower reactivity towards itself due to its relatively large van der Waals volume and a low steric factor [14] of the reaction–$CH_2\bullet + \bullet CH_2$–.

The study of $\varphi$ in different poly(alkyl methacrylates) at different temperatures demonstrated that $\varphi$ depends upon chain relaxation rates and the structure of polymer side chains, but not of free volume $V_f$. ($V_f$ in polymers is usually measured with positron annihilation lifetime spectroscopy (PALS). See ref. [15] for a recent discussion on a free volume in polymers.)

Measurements of $\varphi$ at temperatures in the vicinity of $T_g$ of the studied poly(alkyl methacrylates) led to smooth changes of $\varphi$ vs. $T$ upon crossing $T_g$ [10]. Slight increase of $\varphi$ with temperature was observed for a number of methacrylates [10]. Nanosecond (ns) laser flash photolysis allowed the observation of the fast generation of benzyl radicals after decarbonylation with their subsequent decay. Decay traces demonstrated slow and fast components [9,10], which is expected for geminate recombination in the liquid phase [16,17].

ACOB probe was studied in polyethylene film (PE) of different crystallinities up to $X_c = 68\%$; (B with R= $CH_3$, see Scheme 2) [11]. Crystallinity was found to be one of the factors affecting $\varphi$. "Stiffer" walls of a primary cage (radicals are at $r_c \approx \rho_{12}$) facilitate recombination of radicals. Possibly, rotational diffusion of radicals inside a cage with "stiff" walls leads to their contact by reactive atoms and to recombination in PE. The ordering of RPs of benzylic radicals in the interfacial regions additionally helps the recombination [11]. Continuation of this study [9] demonstrated that $\varphi$ increases at lower temperatures and in PE of higher crystallinity. However, slow relaxation of the walls near $T_g$ leads to lower $\varphi$. It is concluded that shape, $V_f$, wall stiffness, and permeability of the reaction cavities affect $\varphi$. A decrease of the free volume of PE leads to a decrease of the cage escape, as expected [9].

Photochemistry experiments were done in PE not only with ACOB but also with naphthyl esters and ethers undergoing photo-Fries and photo-Claisen reactions [12,13]. The two latter reactions proceed via singlet RPs [12,13]. It was concluded that polymer free volume $V_f$ was the dominant factor controlling the diffusion of the benzyl and other radicals, and, therefore, the kinetics of cage recombination and cage escape [10]. An increase of size (mass) of many atoms radicals leads to a decrease of their cage escape and to an increase of the cage effect [18].

The authors [11] indicate "the versatility of polyethylene as a medium" for the photoinduced reactions. The latter statement can be read as unfortunately poor predictability of PE matrix on the reaction paths.

## 3. Geminate Recombination Kinetics

A series of works [4,19–25] was dedicated to the kinetics of cage effect in polymers or to kinetics of G-pairs decay. Benzophenone (B) was selected as a photoprobe. In general, photoexcited B has been used as a kinetic photoprobe for a long time in many areas—from biochemistry to material science [26]. In the absence of added hydrogen/electron donors, B undergoes photoreduction by polymer matrix with a formation of benzophenone ketyl free radical BH$\bullet$ and a macroradical of a polymer R$\bullet$. Elementary photoinduced reactions of photoreduction of B are presented in Scheme 3:

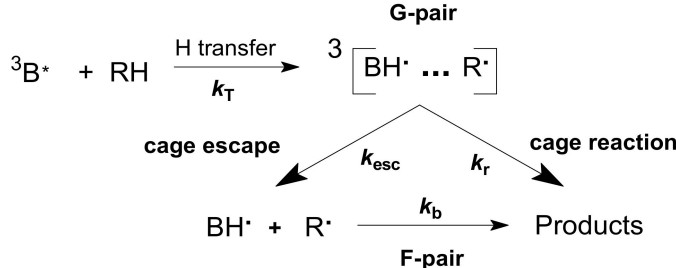

**Scheme 3.** Elementary chemical reactions of photoexcited benzophenone.

Nanosecond laser flash photolysis of B in polymers allows the observation of three processes separated in time: the decay of a triplet state, recombination of G-pairs, and recombination via F-pairs [4]. That makes B a unique and informative photoprobe.

For the first time, kinetics of recombination of G-pairs consisting of BH• and counter radicals were observed in a viscous liquid solution of polymers, namely in trimethysilyl-terminated PDMS elastomer and in 4-(1,1,3,3-tetramethylbutyl)phenyl-polyethylene glycol (a.k.a. Triton X-100) [19].

B-probe was used in the study of soft polymers silicone (PDMS), polystyrene-*block*-poly(ethylene-*ran*-butylene)-*block*-polystyrene (SEBS) (analog of Kraton G1650 of Kraton) and hard polystyrene (PS). The most informative results were obtained in the study of elastomer poly(ethylene-*co*-butylene) films (abbreviated as E) and of stretched polypropylene (abbreviated as BOPP). The rest of the paper is devoted to these two polymers.

Recombination in the polymer bulk proceeds through F-pairs $^{1,3}$[BH• . . . R•]; see Scheme 3.

In publications [20–25], the focus was on a quantitative analysis of kinetic traces of transients, comparison of the kinetics of recombination in the cage and in the polymer bulk (in G- and F-pairs), as well as a study of magnetic field effects (MFEs) on G- and also on F-pairs.

Figure 1 below presents kinetic traces of $^3$B* decay and of geminate recombination at two different temperatures:

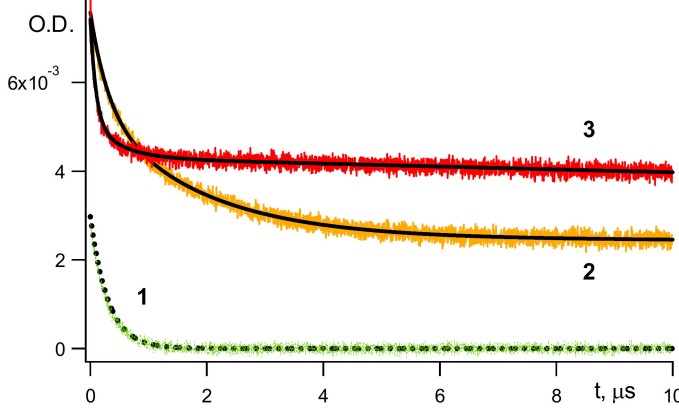

**Figure 1.** Kinetics of $^3$B* measured at λ 630 nm at 263 K obtained under laser flash photolysis of B in E film (1); decay kinetics of BH• measured at λ 545 nm at 263 (2) and at 313 K (3) in the same film. Solid black lines are the computer fit of the experimental curves. In this figure and in figures below, O.D. stands for optical density. Adapted from Ref. [23].

The contribution of $^3$B* into the observed geminate recombination becomes negligible after ~300 μs; see Figure 1.

Cage effect φ by definition is a fraction of BH• decayed in the fast cage reaction (Figure 1(2,3)). φ was calculated by computer simulation decay reactions (Figure 1). Upon completion of a cage decay, a kinetic curve reaches the quasi-plateau; see Figure 1.

It was found that cage recombination/escape in the studied polymers followed the first-order kinetic law. An exponential model of a cage in liquid considers cage recombination and cage escape as competitive reactions of the first order. According to the "simple" exponential model, the cage effect value φ is [4,5]:

$$\varphi = k_r/(k_r + k_{esc}) \tag{6}$$

## 4. Recombination of Radicals in the Bulk of a Polymer

Obviously, cage escape means that radicals enter the solvent bulk and participate in bimolecular recombination in the absence of acceptors. BH•, which is the product of the cage dissociation, decays relatively slowly in the bulk of a polymer (Scheme 3, Figures 2 and 3):

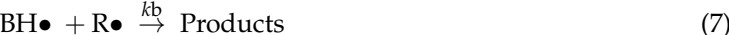

$$BH\bullet + R\bullet \xrightarrow{k_b} Products \tag{7}$$

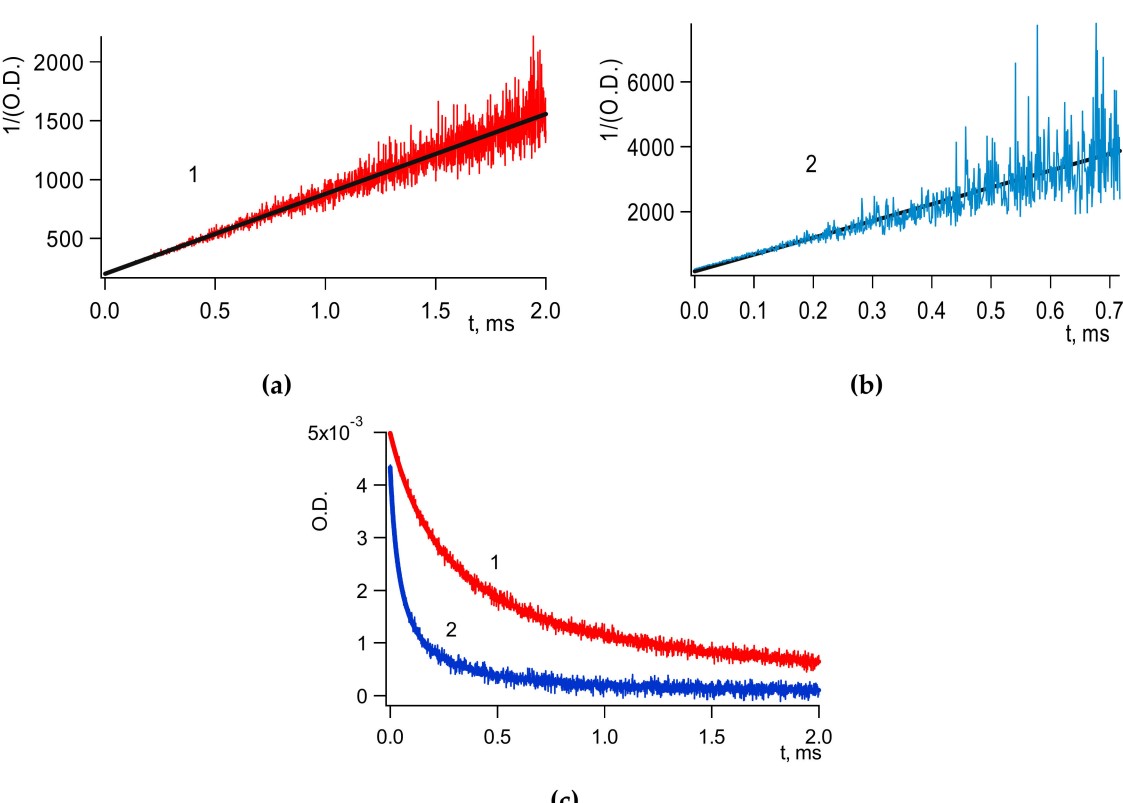

**Figure 2.** Decay kinetics of BH• measured at λ 545 nm at times ≥10 μs obtained under laser flash photolysis of B in E films at 263 (1) and at 313 K (2). The top of the figure: A fit of curves into the second-order kinetic law. Adapted from Ref. [23]. (**a**) A fit of curve 1 of the figure c; (**b**) A fit curve of curve 2 of the figure c; (**c**) Experimental kinetic curves.

Scheme 3 demonstrates that radicals are produced as pairs. Obviously, initial concentrations of the radicals in the bulk of a polymer are the same: [R•]o = [BH•]o. Figure 2 presents a good fit of the kinetics of BH• decay as the second-order kinetics. Second-order rate constants $k_b$ for several polymers were reported [22–25].

Production of benzpinacol is the chief route of BH• recombination in liquids [2]. However, this route of BH• decay is apparently minor in polymers. R• is a slowly moving dormant radical. In such cases, a cross-recombination of radicals is the key path of their decay (the persistent radical effect [27]). This result was observed also in a reaction of benzyl radical with poorly reactive 4-hexadecylbenzyl radical [9] (see above).

There could be many methods of possible quantitative study of kinetics in polymers. One of them is dispersive kinetics (see e.g., Ref. [28]). We used a function of Ref. [28] in the examination of decay kinetics of $^3$B*, F-and G-pairs for the kinetics of transient decays in hard polymers [20,21]. At the same time, "simple" first-and second-order kinetics satisfactorily defines corresponding reactions in the studied polymers [4,23–25].

Recombination of G-pairs and F-pairs was studied in several poly(urethane acrylates) and polyacrylate polymers [20]. It was found that an increase of hardness of a polymer measured by durometer A and by other parameters characterizing hardness leads to an increase of φ and retards recombination in the polymer bulk [20].

## 5. Magnetic Field Effects on the Reaction Kinetics in Geminate and Free-Radical Pairs

Usage of magnetic field (MF) slows down recombination of a triplet geminate pair and recombination of radicals in a bulk of a viscous solvent or a polymer [3,4,29]. Figures 3 and 4 below demonstrate a couple of examples.

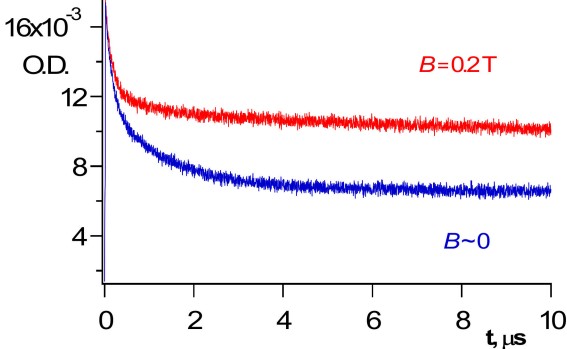

**Figure 3.** Kinetic traces of transients at λ 545 nm measured in E. BH• decay was measured within G-pairs (Scheme 3) in the Earth's MF and in the external MF. Adapted from Ref. [24]; see this Ref. for details.

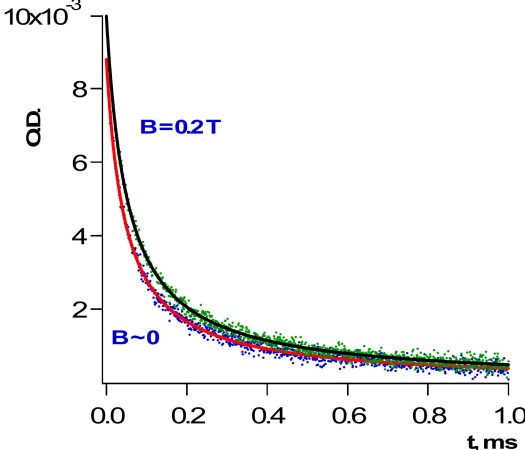

**Figure 4.** Kinetic traces of BH• decay measured at λ 545 nm. They correspond to the second-order reaction (7) in E proceeding in the external MF and in the Earth's MF. Solid lines present computer simulation of experimental data (dots) into the second-order kinetics. Adapted from Ref. [24]; see this Ref. for details.

Experimental data on the measured and calculated rate constants and on φ in the Earth's and in the external MF of 0.2 T are presented in Refs. [4,22–24]. In particular, in an elastomer (E), a decrease of φ more than two times and a decrease of $k_b$ (Equation (7)) by ~20% was observed in MF [24].

Within the exponential model, $k_{esc}$ is a parting of radicals, which should not depend on MF. In fact, no MFE on $k_{esc}$ within an accuracy of $k_{esc}$ determination was observed [4]. Significant MFEs are observed in a media of high but optimal viscosity [3], and apparently many studied polymers are such a media.

The probability σ of recombination via F-pairs in the absence or in the presence of MF is [4,29]:

$$\sigma = 0.75 \cdot \varphi + 0.25, \tag{8}$$

where 0.75 is the probability of an encounter of the pair in a T-state and 0.25 is the probability of an encounter of the same pair in an S-state [29]. Further [14],

$$k_b = \sigma \cdot k_{diff}, \tag{9}$$

where $k_{diff}$ is a rate constant of a diffusion-controlled reaction (rate constant of double encounters).

It is assumed that S-pairs react with a probability 1.0, T-pairs react with a probability φ measured in the external MF or in Earth's MF. It is obvious to accept that F-pairs in a T-state have the same reactivity as G-pairs, supposing no significant steric limitation on the reaction [3,14]. (Radicals in triplet F-pairs encounter each other in a random mutual orientation, whereas photogenerated triplet G-pairs maybe formed in a certain specific mutual orientation [6].)

Experimental MFEs on $k_b$ (Equation (7)) coincide within their determination error with the related MFEs calculated with Equation (8) [23,24].

The most probable mechanism of the observed MFE in E [22–24] is the hyperfine coupling (HFC) mechanism [29]. Briefly, the HFC mechanism acts the following way. The RPs exist in an S (total electron spin is zero) or T (total spin is 1.0) spin state; see Equation (3). Only S-state leads to reaction products. Magnetic hyperfine interactions with nuclear spins or applied MF may affect the reactivity of RPs by altering the spin angular momentum of an RP (S or T); see Refs. [2,29] for details.

## 6. Photoreduction of B in the Stretched Elastomer E

Elastomers are often stretched during their use. An elastomer package holding one or another load or rubbery O-rings during its use are common examples. Photoprobe B is expected to give info on changes in the studied E under stretching [24]. Films of E with a dissolved B or a dissolved stable nitroxyl radical (spin probe) had initial thickness $\zeta_o = 0.007$–$0.06$ cm. Films were stretched up to three times, and the thickness of the stretched films was measured and used in the study [24].

It was assumed that $k_b$ (Equations (7) and (9)), as well as $k_{esc}$ (Scheme 3) are directly proportional to a coefficient of translational diffusion of BH• $D_{BH}$ [3,22,23]:

$$k_b(k_{esc}) \sim D_{BH} \tag{10}$$

(Macroradical R• should be practically immobile.) Experimentally determined $k_{esc}$ and $k_b$ essentially depend upon stretching; see Figure 5 [24].

A valuable observation is that reaction (7) proceeds slower in a stretched polymer than in non stretched polymer of the same thickness $\zeta$, as presented in Figure 5.

Diffusion in polymers is a rather complicated phenomenon (see Section 2 above). It depends on diverse properties of a diffusant and of the polymer [15]. An important diffusant property is its size/dimensions. Within the free volume ($V_f$) theory by Vrentas–Duda, the diffusion of low-MW molecules in polymers essentially depends on $V_f$ of a polymer [15,30]. A linear dependence of $\log_{10} D_{BH}$ upon $1/V_f$ is expected within the framework of $V_f$ theories [15,30].

So, it is possible to assume:

$$\log_{10} D_{BH} = a - b/V_f, \tag{11}$$

where $a$ and $b$ are constants [15,24,30].

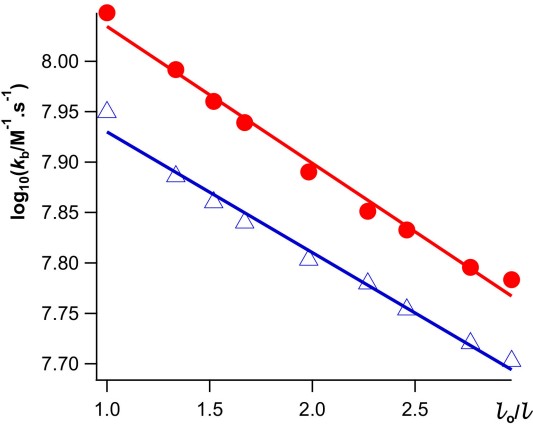

**Figure 5.** Dependencies of $\log_{10}k_b$ upon stretching of films in the Earth's MF (circles) and in MF $B = 0.2$ T (triangles). $\zeta_o$ was different for the studied E films. Adapted from Ref. [24].

It is presumed that stretching of E leads to a decrease of $V_f$, to a decrease of density of holes/pores which are required for the translational diffusion. (Such a decrease does not influence the rotational diffusion of a spin probe in E [24].) In addition, segmental movements of a polymer chain assist diffusion of low-MW molecules. We need to assume that stretched chains lose that facility to some extent.

Another conjecture is that the dimensionless fractional $V_f$ [24]:

$$V_f \sim \zeta/\zeta_o \qquad (12)$$

Combining Equations (10)–(12), one can obtain:

$$\log_{10}k_b \ (\log_{10}k_{esc}) = \text{const} - b \cdot \zeta_o/\zeta \qquad (13)$$

where $b \approx 0.13$ for both escape (Scheme 3) and recombination in the solvent bulk (7). It seems that $b$ is a characteristic of E. Linear dependence (13) for $k_b$ was presented in Figure 5.

Application of moderate MF leads to retardation of cage escape (Scheme 3), reaction (7), and $\varphi$-value (Equation (6)); see Figures 3–5. Stretching of E does not affect MFE.

## 7. Photoreduction of B in the Stretched Polypropylene BOPP

Study of (stretched) elastomers with B probe gives valuable information on elementary reactions in such media; see Sections 3–6 above. This section is devoted to the study of biaxially oriented polypropylene (BOPP) with B probe. BOPP is a valuable polymer which is predominantly used in packaging. "Biaxially oriented" signifies that the polypropylene (PP) film was stretched in two directions perpendicular to each other. BOPP has oriented chains, high modulus, high stiffness, and certain crystallinity due to its elongation during production [31]. That makes a semicrystalline BOPP an interesting object for investigation with B probe. Investigation of photoinduced reactions in BOPP preheated up to its melting point will provide information on the stability of BOPP at elevated temperatures. The study of properties of BOPP during its thermal treatment was performed after BOPP preheating for a short time in a temperature range of 298–423 K. These temperatures are much lower than BOPP melting point of 453 K. All results with preheated (annealed) and with non- preheated BOPP were obtained at 298 K. Properties of BOPP containing B were studied with ESR (nitroxyl spin probe), WAXD, DSC, and IR [25].

Elementary reactions observed under photoexcitation of B in BOPP are presented on the already-known Scheme 3; see above. However, no exit radicals in the solvent bulk were

observed in non-preheated BOPP [25]. RPs probably decay as pairs in close contact $^3$[BH•, R•] in non-preheated polymer:

$$^3[\text{BH•, R•}] \rightarrow \text{Products} \tag{14}$$

A critical phenomenon takes place: dissociation of RP with a formation of free radicals in the polymer bulk is observed at preheating temperature $T_{\text{crit}} \approx 403$ K and at a higher $T$; see Figure 6 [25]:

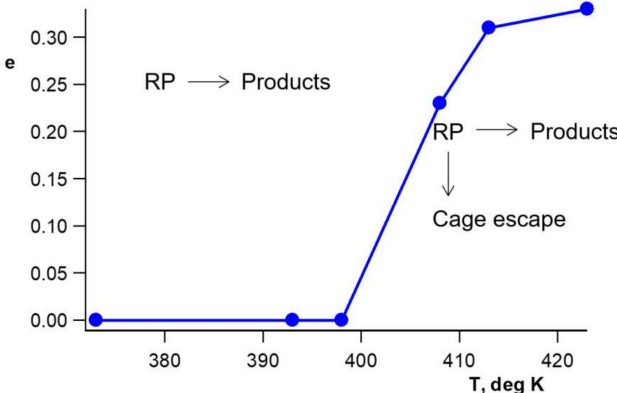

**Figure 6.** Plot of cage escape value *e* obtained at 298 K in B@BOPP films upon preheating temperature. Adapted from Ref. [25].

The physical process of heating and cooling of BOPP most probably leads to the restructuring of crystallites, shrinking of the distribution of crystallites according to their sizes in BOPP, and agglomeration of crystallites. BOPP becomes softer, which manifests itself in the observed kinetics of BH• decay. BH•, which enters the polymer bulk at the preheating $T \geq 403$ K, decays according to Equation (7). Enclosed in the soft amorphous phase of BOPP, radicals BH• and the counter radicals R• are in a relatively high concentration. That leads to a high observed rate constant $k_b \sim 10^8$ M$^{-1} \cdot$s$^{-1}$ for this reaction [25].

Scheme 4 below presents changes in B@BOPP occurring at $T \approx 403$K:

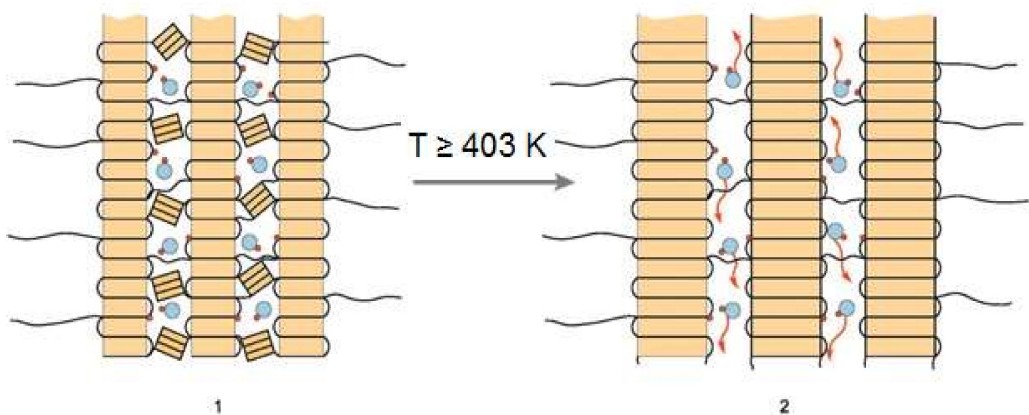

**Scheme 4.** Geminate reactions in B@BOPP at 298 K (case 1) and at $T \geq 403$ K (case 2) as a pictorial presentation. Blue circle with a red dot stands for BH•. An unpaired electron of R• and BH• is presented by a red dot. BH• and its counter radical R• are photogenerated in proximity to each other in cases 1 and 2. Individual polymer molecules are presented as wavy lines. In the case 2, BH• can exit the RP. See details in Ref. [25].

The crystallites are assumed to be large, flat, and situated parallel to each other. There is an amorphous phase of BOPP in the room between crystallites. The crystallites do not stick to each other; there is an amorphous phase between crystallites without preheating (case 1). That amorphous phase

has many relatively small crystallites in accordance with the data obtained by DSC [25]. Most probably small crystallites are created due to mechanical damage of the whole crystal phase of PP in the course of manufacturing of BOPP.

According to the ESR data with a spin probe, the amorphous phase (case 1) is in a strained and less mobile state compared to case 2 (Scheme 4). BOPP (case 1) is a typical polymer with barriers for a retarded diffusion. Such a condition limits the mobility of BH•. In case 2, the amorphous phase relaxes due to preheating and allows faster translational diffusion of BH• compared to case 1 [25]. Almost all small crystallites merge with large crystallites at $T$ close to 423 K [31]. In estimation of ref. [25], that merger leads to an increase of the average size of crystallite by 1.5 times.

BOPP demonstrates approximately the same total percent of crystallinity $X_c \approx 45\%$ in cases 1 and 2 [25]. Case 1 in Scheme 4 demonstrates that the environment of BH• obstructs its exit into the polymer bulk ($e = 0$, $\varphi = 1.0$). However, in case 2, an entry into the polymer bulk is possible ($e > 0$); see Scheme 2. Possible directions of the BH• cage escape are depicted by red wavy lines in Scheme 4, case 2.

This result—independence of $\varphi$ vs. $X_c$ [25]—differs from an observation on PE [9], where an increase of $\varphi$ was related to the increase of $X_c$ (see Section 2 above). According to our assumption, the size of crystallites changes with temperature despite minor changes in the observed $X_c$. Scheme 4 reflects this statement [25].

It is important to note that such a widely used technique in polymer research as WAXD demonstrates only increase of crystallite size in BOPP after preheating. At the same time, kinetic results demonstrate a critical phenomenon in the temperature behavior of BOPP cages.

## 8. Conclusions

Cage effects are observed during photolysis, thermolysis, radiolysis, sonolysis, and in gases under high pressure. In this review article, we considered only cage effect studies under photolysis of low-MW compounds dissolved or distributed in a liquid or solid under ambient conditions polymer. The article was devoted mainly to the study of reactions of free radicals studied by two probes, ACOB and B, in the polymer bulk and in a polymer cage.

An exponential model of a cage effect in elastomers nicely describes the experimental kinetics [4]. External MF affects polymer free-radical reactions. It was established for the first time that triplet F-pairs have quite similar reactivity as triplet G-pairs [4].

It is hard to make conclusions on the effect of one or another property of polymer on $\varphi$ (e) due to the diversity of polymers' chemical and physical properties. However, one can start with the analysis of polymer free volume $V_f$. The following correlation is often observed for solid polymers: cage escape rate $k_{esc}$ and value $e$ usually increase with the $V_f$ - increase.

Free radicals are reactive species, and their self-termination is most probably limited by diffusion in viscous polymer matrices. Diffusion in polymers depends on many factors. The expected tendency is that diffusion depends upon $V_f$, and Vrentas–Duda theory [30] provides a good approach.

Nitroxyl radicals have been used for a very long time for a study structure of polymers [32]. ESR data on nitroxyls in polymers provides information on a rotational diffusion of probes in a polymer [25,32]. Presented in this paper, investigation of the kinetics of BH• supplies data on a translational diffusion in a polymer of closely positioned radicals (G-pairs) and of separated free radicals (F-pairs) [19–25].

There are a number of problems related to cage effect in the condensed phase and in polymers in particular. Theoretical study of cage effect dynamics is an area of chemical physics (e.g., Refs. [17,33]). The molecular structure of a liquid or of a polymer should be taken into account under consideration of diffusion of reagents at a distance of few $\rho_{12}$ [6]. A promising approach is using quantitative values of "microviscosity" affecting $\varphi$ [34]. It is important to learn where guest molecules are located in a polymer. Solid-state NMR spectroscopy should be a useful method to determine location of probes.

**Funding:** This research received no external funding.

**Conflicts of Interest:** The authors declare no conflict of interest.

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
