# Peer review of "Cage Effect under Photolysis in Polymer Matrices"

_coatings, doi:10.3390/coatings9020111_

Reviewer 1 Report

This review article summarizes the cage effect of radical pairs generated from precursors by photolysis in polymer matrices studied mainly by the authors’ group. As the authors concluded as “it is hard to make one-two solid conclusions on effect of …”, the manuscript was not intended to provide a take home message to the readers but rather to show collections of several experimental results obtained by the authors’ group. Despite the lack of clear conclusion, the manuscript would be useful for readers working in this field. As this is a review article, I am not going to discuss chemistry but rather ask to collect formalism.

Page 2, line 41-: Please add references for the origin of Equation 1. Also, please provide dimensions of each parameters; rc, tc, r, and D.

Page 2, line 45-: Definition of the solvent cage is wrong. The in-cage reaction is unaffected by even scavengers. Otherwise one cannot rigorously distinguish the in-cage and out-of-cage reactions, such as the reaction shown in Scheme 2.

Page 3, line 89: Please provide more detailed explanation why the spin-orbit coupling enables the coupling of a triplet pair.

Pages 4 and 11: In the case of PE matrix, the authors concluded that the crystalline fraction just serves as a “stiff” wall and reduce polymer free volume to explain the observed cage effect (page 4). However, in the case of stretched BOPP, the authors mentioned that the change in the crystalline size could not explain the observed cage effect. Why were complete different conclusions reduced to the same phenomenon?

Page 7, Figures 3 and 4: It is unclear the difference between Figure 3 and 4.

Page 8, line 238-: Please provide more detailed explanation about the hyperfine coupling mechanism.

Author Response

General Comment: This review article summarizes the cage effect of radical pairs generated from precursors by photolysis in polymer matrices studied mainly by the authors’ group. As the authors concluded as “it is hard to make one-two solid conclusions on effect of …”, the manuscript was not intended to provide a take home message to the readers but rather to show collections of several experimental results obtained by the authors’ group. Despite the lack of clear conclusion, the manuscript would be useful for readers working in this field. As this is a review article, I am not going to discuss chemistry but rather ask

Authors’ feedback and revision: It would be strange to prepare a review article based predominantly on publications of other researchers. There are not many publications on the subject to the best of our knowledge. Our group pioneered observation of the cage effect dynamics in polymers, see ref. 19 (1987). We cited practically all of the publications of Weiss, Chesta and their co-authors who worked in this area as well, see refs 3, 9-13. It is hard to make a clear conclusion in this diverse area. Still, we revised the Conclusion section in order to meet a Reviewer’s comment, see lines 357-360.

Comment: Page 2, line 41-: Please add references for the origin of Equation 1. Also, please provide dimensions of each parameters; rc, tc, r, and D.

Authors’ feedback and revision:  The references are added, see line 44. Dimensions are provided. See lines 41-44.

Comment: Page 2, line 45-: Definition of the solvent cage is wrong. The in-cage reaction is unaffected by even scavengers. Otherwise one cannot rigorously distinguish the in-cage and out-of-cage reactions, such as the reaction shown in Scheme 2.

Authors’ feedback and revision: Usually scavengers are added to a solution in a concentration of the order of mM. Scavenger added in a concentration of ~1M and higher becomes a cosolvent. In such rare case scavenger molecule(s) are located in the wall of a primary cage and react with radicals (see ref. 5 and an old publication: Koenig, T.; Fischer, in Free Radicals; Kochi, J., Ed.; John Wiley: New York, 1973; vol.1,ch 4.)  Still, we changed the line 46 and added ref.5 on this line.

Comment: Page 3, line 89: Please provide more detailed explanation why the spin-orbit coupling enables the coupling of a triplet pair.

Authors’ feedback and revision: Done. See lines 92-95.

Comment: Pages 4 and 11: In the case of PE matrix, the authors concluded that the crystalline fraction just serves as a “stiff” wall and reduce polymer free volume to explain the observed cage effect (page 4). However, in the case of stretched BOPP, the authors mentioned that the change in the crystalline size could not explain the observed cage effect. Why were complete different conclusions reduced to the same phenomenon?

Authors’ feedback and revision: Experiments in PE matrix were described in publications Weiss and co-authors in refs 9,11-13, see lines 123-137. They suggested in particular that “stiff” walls lead to a higher cage effect. A higher crystallinity of PE at lower temperatures leads to the same result. In our work on BOPP (ref. 25) the total crystallinity practically did not change with temperature (lines 336), but the form of crystals did change with temperature, see Scheme 4. We added an additional clarification of this subject on lines 340-343.

Comment: Page 7, Figures 3 and 4: It is unclear the difference between Figure 3 and 4.

Authors’ feedback and revision: Both Figures present kinetics. Figure 3 presents kinetics in the microsecond time scale, Figure 4 presents kinetics in the millisecond time scale. Figure 3 demonstrates the effect of external magnetic field (MF) on G-pairs, whereas the data of Figure 4 demonstrates the effect of MF on F-pairs. See the Figures and the relevant captions.

Comment: Page 8, line 238-: Please provide more detailed explanation about the hyperfine coupling mechanism.

Authors’ feedback and revision: Done. See lines 24-250.

Reviewer 2 Report

Though this MS would fit better in Journal with focus on photochemistry or physical chemistry, it also complements the themes published in a journal with focus on coatings. The content is interesting and helps to get an understanding about occurring reactions in polymers. The MS is well organized but there are some changes necessary:

1) Graphics: the proportions of some graphic elements in schemes and figures does not look professional. This relates to line thickness, fonts sizes of axis and symbol size. This needs improvement. I believe the journal give some help.

2) The ms reads sometimes like a better lab report. I would expect a better discussion of the results obtained. I recommend that authors run through it and make improvements accordingly. For example, the parts relating to the stretching and magnetic field needs more attention. The results obtained were sold under value. This has much more potential. Perhaps authors may also have more experimental data that can be included.

Overall I recommend publication after completing the requested changes.

Author Response

General Comment: Though this MS would fit better in Journal with focus on photochemistry or physical chemistry, it also complements the themes published in a journal with focus on coatings. The content is interesting and helps to get an understanding about occurring reactions in polymers. The MS is well organized but there are some changes necessary:

 Authors’ feedback and revision: We agree with a Reviewer. However, most of the organic coatings produced by photopolymerization have a residual low MW photoinitiator (PI). That PI becomes a cause of photodegradation of the cured coatings outdoors. In particular, benzophenone, which we studied, is a common PI. Thus, we believe that a manuscript will be useful to coatings researchers and chemists. We added these considerations in lines 72-75.

Comment: 1) Graphics: the proportions of some graphic elements in schemes and figures does not look professional. This relates to line thickness, fonts sizes of axis and symbol size. This needs improvement. I believe the journal give some help.

Authors’ feedback and revision: We improved where we could presentation of Schemes and Figures, see the text. At the same time the Figures/graphics were prepared professionally with software IgorPro 3.14 and ChemDraw. We had to demonstrate the on the kinetics curves the original experimental points; line thickness reflects the determination error. We would appreciate “some help” of the Journal if necessary as the Reviewer suggested.

Comment: 2) The ms reads sometimes like a better lab report. I would expect a better discussion of the results obtained. I recommend that authors run through it and make improvements accordingly. For example, the parts relating to the stretching and magnetic field needs more attention. The results obtained were sold under value. This has much more potential. Perhaps authors may also have more experimental data that can be included.

Overall I recommend publication after completing the requested changes.

Authors’ feedback and revision: We made attempts to improve the manuscript in line with the Reviewer’s comment. At the same time, the manuscript is hardly a lab report. It starts with the first publication on the cage effect; it gives a broad introduction into cage effect problems in the Introduction and in the Conclusion Sections. Unfortunately, we do not have many additional data deserving publication on “stretching” and on “magnetic field”. We did not want to go into more details here in order not to expand a review article and bury the reader in details. Corrections and additions made per another Reviewer’s comments provide additional clarification into the observed magnetic field effects.  More details on all the subjects can be found in the referenced original publications.

We are pleased that the Reviewer overall recommends publication.

Round  2

Reviewer 1 Report

The authors modified several important issues so that the manuscript is publishable as it is.

Reviewer 2 Report

The content of this paper is interesting for coatings seeing light in their life. This can be either sunlight or processing light used for curing. The results obtained help to understand the behavior of radicals and the interaction of light active materials in coatings.